# Enhancing Cricket Farming Efficiency: Comparative Analysis of Temperature Effects on Three Edible Malagasy Species

**DOI:** 10.3390/insects16090960

**Published:** 2025-09-12

**Authors:** Valéry Riantsoa, Henlay J. O. Magara, Sylvain Hugel, Brian L. Fisher

**Affiliations:** 1Department of Feed Development, Madagascar Biodiversity Center, Antananarivo 101, Madagascar; mhenlay@gmail.com (H.J.O.M.); hugels@inci-cnrs.unistra.fr (S.H.); bfisher@calacademy.org (B.L.F.); 2Department of Entomology, University of Antananarivo, Antananarivo 101, Madagascar; 3Institut des Neurosciences Cellulaires et Intégratives, UPR 3212, CNRS–Université de Strasbourg, 67000 Strasbourg, France; 4Department of Entomology, California Academy of Sciences, San Francisco, CA 94118, USA

**Keywords:** edible crickets, thermal response, life-history traits, Malagasy crickets, cricket farming

## Abstract

**Simple Summary:**

As the world seeks sustainable sources of protein, edible crickets present a viable solution to food shortages and environmental challenges, particularly in Madagascar. This study examined how two different temperatures, 28 °C and 30 °C, affect the growth and survival of three cricket species: *Gryllus bimaculatus*, *Gryllodes sigillatus*, and *Teleogryllus lemur*. We found that *Gryllus bimaculatus* thrived at the higher temperature, growing faster and producing more biomass without losing many individuals. In contrast, *Gryllodes sigillatus* maintained good survival rates but grew more slowly, while *Teleogryllus lemur* showed significant growth but suffered high mortality at the warmer temperature. These results highlight the importance of choosing the right species and temperature for cricket farming to maximize production while ensuring survival. For farmers, especially those using climate control, raising temperatures slightly could enhance productivity, but careful consideration of costs and species sensitivity is essential. This research provides valuable insights for developing effective and sustainable cricket farming practices, which could help address food security and promote environmentally friendly protein sources in tropical regions.

**Abstract:**

As demand for sustainable protein intensifies, edible crickets offer a promising solution to food insecurity and environmental strain, particularly in regions like Madagascar. This study investigated the effects of two rearing temperatures (28 °C and 30 °C) on life-history traits of three edible cricket species—*Gryllus bimaculatus*, *Gryllodes sigillatus*, and *Teleogryllus lemur*—under controlled laboratory conditions. Growth, survival, development time, and biomass yield were quantified for each species (*n* = 150 per temperature treatment) and analyzed using *t*-tests, chi-squared tests, and ANOVA. Results revealed distinct species-specific thermal responses. *G. bimaculatus* exhibited accelerated development and higher biomass at 30 °C without significant survival loss. *G. sigillatus* maintained high survival but developed more slowly at 30 °C, suggesting thermal sensitivity. In contrast, *T. lemur* demonstrated substantial growth gains at 30 °C but suffered a 50% reduction in survival, indicating heat intolerance. Development time and biomass yield varied significantly across species and temperature treatments (*p* < 0.0001). These findings highlight critical trade-offs between growth efficiency and survival in insect farming systems. *G. bimaculatus* emerges as a strong candidate for high-yield, thermally resilient farming, while *T. lemur* requires cooler rearing environments. This work informs species selection and environmental optimization for scalable, climate-adaptive cricket farming in tropical regions.

## 1. Introduction

As global populations grow and conventional food systems strain under environmental and economic pressures, the need for sustainable, nutrient-dense protein sources is no longer a future ambition but an urgent necessity. Edible insects, long consumed in traditional diets worldwide, are increasingly recognized as viable “mini-livestock” capable of addressing food insecurity and ecological sustainability [1]. Among them, crickets stand out for their exceptional nutritional quality, efficient feed conversion, and low environmental footprint [2,3]. Their ability to thrive on agro-industrial by-products and wild plants, as well as their adaptability to controlled farming conditions, makes them ideal candidates for scalable insect farming systems [4,5].

Cricket farming now takes place across very different scales, from household-level rearing in rural communities to large, automated facilities in Asia, Europe, and North America [6]. Although overall production is still small compared to conventional livestock, it is growing quickly as new technologies and investments emerge [1,2,3,4,5,6]. In addition to human food, crickets are also farmed for feed, aquaculture, and pet markets, showing how the sector is moving from a niche practice toward a more structured agro-industry with global relevance [6]. Crickets are rich in high-quality protein, essential amino acids, vitamins, minerals, and chitin, offering a promising solution to micronutrient deficiencies in low-resource settings [7,8,9]. Compared to conventional livestock, they require significantly less land, water, and feed, and produce far fewer greenhouse gas emissions [10]. These advantages, combined with the ability to rear them on local waste streams, enhance their potential in decentralized and sustainable food systems [6,7,8,9,10,11].

In Madagascar, where over 70% of the population experiences food insecurity and nearly half of children under five are chronically malnourished [12], cricket farming is emerging as a culturally compatible and scalable solution. Native edible species such as *Gryllus bimaculatus* De Geer, 1773 have demonstrated rapid development, high protein content, and low production costs when raised on locally available feed inputs [13,14]. Crickets are already part of traditional Malagasy diets [15], and by-products like cricket frass have been shown to enhance crop production, integrating insect farming further into sustainable agricultural practices [16].

To develop resilient and productive farming systems, it is essential to optimize both feed quality and environmental conditions. Among these, temperature plays a pivotal role in shaping the biology and yield of ectothermic insects. In crickets, temperature influences development rate, growth, final body mass, fecundity, and survival [17,18,19,20]. For example, *Gryllus bimaculatus* raised at 30–32 °C exhibits faster growth, higher biomass, and greater fecundity compared to those reared at 28 °C, where development slows and mortality can increase [13,21].

While 28 °C is frequently used in experimental rearing for its balance between survival and developmental rate [22,23], commercial operations often target 30 °C to accelerate production. This temperature window (28–30 °C) is generally considered optimal for species like *Acheta domesticus* (Linnaeus, 1758) and *Gryllodes sigillatus* (Walker, 1869), though performance responses can vary depending on species and strain [24,25]. Moreover, traits such as body size and development time are directly linked to fecundity and total yield [26,27], and thus are critical for evaluating species’ suitability for farming.

Despite the expanding literature on cricket farming, *Teleogryllus lemur* Gorochov, 1990 remains largely underexplored, and comparative studies among edible species in Madagascar are rare. Although data exist for *G. bimaculatus*, *A. domesticus*, and *G. sigillatus* in other global regions [28,29], their relevance under local Malagasy conditions is not well understood. Furthermore, detailed assessments of temperature effects on life-history traits are critical for scaling insect production systems reliably in tropical regions.

This study aimed to evaluate the effects of two commonly used rearing temperatures—28 °C and 30 °C—on growth, survival, development time, and biomass yield in three edible cricket species: *Gryllus bimaculatus*; *Gryllodes sigillatus* and *Teleogryllus lemur* These species were selected for their farming potential, native distribution in Madagascar, and contrasting life-history traits. By generating comparative data under controlled conditions, this research aims to inform species selection, refine rearing protocols, and contribute to the development of sustainable insect farming systems adapted to Madagascar and similar tropical environments.

## 2. Materials and Methods

### 2.1. Cricket Species and Rearing Conditions

Three species of crickets used in this study were collected from three distinct regions in Madagascar: Fort Dauphin, Morondava, and Mahajanga (Table 1). The crickets included *Gryllus bimaculatus* De Geer, 1773; *Teleogryllus lemur* Gorochov, 1990; and *Gryllodes sigillatus* (Walker, 1869). In the laboratory at the Madagascar Biodiversity Center, crickets were reared under controlled conditions of 28 ± 1 °C and 60 ± 5% relative humidity. Conditions were monitored daily using a digital thermohygrometer. Rearing groups were maintained for 6 generations to build enough individuals before experimentation. Crickets were reared in ventilated plastic containers (50 cm × 35 cm × 30 cm), with a density of 200 individuals per container. Each container contained four egg carton trays for shelter. A 12:12 h light–dark photoperiod was maintained, and crickets were given *ad libitum* access to food and water. The diet consisted of standard chicken feed sourced from a local supplier, provided uniformly across all species. Cotton balls were provided as oviposition substrates for adult crickets.

### 2.2. Temperature and Diet

Two temperature conditions, 28 °C and 30 °C, were selected for their influence on developmental time and survival, with 30 °C known to reduce developmental duration [30] and support growth [20]. Temperature 28 °C was selected since most of the reared insects have been reported to grow well at this temperature. Previous research on *Teleogryllus emma* (Ohmachi & Matsuura, 1951), demonstrated that 28 °C enhances survival [23].

A commercial chicken feed was used for both the experimental diet and colony maintenance. The feed was procured from a certified local supplier (LFL Madagascar, Antananarivo, Madagascar), and its proximate composition was analyzed by the Nutritional Laboratory of FOFIFA Madagascar. The commercial chicken feeds used at 28 °C (18.2% CP) and 30 °C (21.5% CP) were sourced locally from different production batches. Although these values differ slightly, both fall within the typical dietary protein range for optimal cricket rearing (18–22% CP). This variability reflects practical conditions farmers face when relying on local feed sources. The diet contained the following nutrient ranges: at 30 °C, crude protein (CP) [21.5%], moisture [12.5%], fat [3.1%], ash [2.1%], fiber [4.0%], and available carbohydrate [40.1%]; at 28 °C, crude protein (CP) [18.2%], moisture [12.2%], fat [3.1%], ash [6.4%], fiber [5.0%], and available carbohydrate [40.3%].

### 2.3. Experimental Design

Eggs for the experiment were obtained from the mother colony by providing cotton balls as oviposition substrates for two hours. After two hours, the cotton balls were removed from the colony, opened into cotton sheets for easy removal of the eggs. From the cotton ball, 300 eggs detached and counted. The counted eggs were introduced into wet fresh cotton sheets. The cotton sheets with the eggs were covered with another thin layer of wet cotton sheet. A total of 300 eggs per species were incubated in ventilated boxes measuring 15 cm × 10 cm × 10 cm, kept moist with daily water sprays. Following hatching, 150 individual crickets from each species were reared under each temperature condition (30 °C or 28 °C), divided into three replicates of 50 individuals, each housed in a separate incubator (PHCBI^®^ MIR-554, PHC Corporation, Tokyo, Japan). Within each replicate, crickets were placed in 50 rearing boxes measuring 15 cm × 10 cm × 10 cm, randomly arranged to minimize positional bias and ensure robust statistical reliability. The rearing conditions were maintained constant in controlled incubators, with relative humidity set at 70 ± 5% and a 12:12 light–dark photoperiod. Temperature and humidity levels were monitored using a data logger. Each cricket was provided with *ad libitum* food and water, and moist cotton balls were replaced every three days. Egg cartons measuring 4 cm × 4 cm offered shelter to crickets in each container.

### 2.4. Data Collection

#### 2.4.1. Growth Measurement

Body length and weight were measured weekly using a caliper (Fisher Darex^®^ 150 mm, Fischer Darex, Chambon-Feugerolles, France) with a precision of 0.01 mm, and an electronic balance (Digital Cara Scale^®^ DS-04B, 20 g, Hanyu Electronic Technology Co., Ltd., Shenzhen, China) with a precision of 0.001 g. Crickets were placed in Petri dishes during measurement.

#### 2.4.2. Survival Rate

Survival was monitored weekly by counting dead individuals. The total number of surviving crickets was recorded according to methods by Miech et al. [31].

#### 2.4.3. Development Time

Developmental time was defined as the period from hatching to adulthood, indicated by the last molting event. The date of the last molt was recorded to measure this duration. Molting timings were recorded by inspecting the containers every three days for exoskeletons, which indicated recent molting. Exoskeletons were collected and examined on white paper. To assess the result of development time the data was taken from surviving individuals till the last molting event.

#### 2.4.4. Biomass Yield

Biomass yield was calculated by dividing the total weight of surviving adults by their development time at each temperature.

### 2.5. Statistical Analyses

To study the effect of temperature on growth, survival, development time, and biomass yield across the three cricket species, simple and clear statistical methods were used.

#### 2.5.1. Body Length and Body Weight

The differences in body length and body weight between temperatures (28 °C and 30 °C) were tested using *t*-tests for each species. When the groups did not have equal variances, Welch’s *t*-test was used. A result was considered significant if *p* was less than 0.05.

#### 2.5.2. Survival

Survival rates were analyzed by creating weekly survival curves for each species at both temperatures. Final survival was calculated as the percentage of individuals still alive at the end of the study, based on an initial group of 150 crickets. To find out if survival differed between the two temperatures, a chi-squared test (χ^2^ test) was used.

#### 2.5.3. Development Time

A two-way analysis of variance (ANOVA) was used to look at the effects of temperature, species, and their interaction on development time. Levene’s test was done to check if the variance between groups was equal. If it was not equal, Welch’s *t*-tests were used to compare development times for each species at 28 °C and 30 °C. Results were reported as averages with standard errors (mean ± SE), and differences were considered significant if *p* was less than 0.05.

#### 2.5.4. Biomass Yield

Biomass yield (g/day) per cohort was calculated for each species and temperature treatment by dividing the total final body weight of surviving individuals by the mean development time (days). To test the effect of temperature on biomass yield within each species, one-way ANOVA was conducted separately for *Gryllodes sigillatus*, *Gryllus bimaculatus*, and *Teleogryllus lemur*. Significance was assessed at *p* < 0.05.

#### 2.5.5. Tools and Software

All tests and graphs were done using R version 4.4.2 (Core Team) (2024) statistical software. The ‘dplyr’ package helped process the data, ‘ggplot2’ was used for graphs, and the ‘stats’ package was used for the tests. Results were shown as test values (t, F, or χ^2^), degrees of freedom, and *p*-values to show significance. The threshold of statistical significance was set at *p* < 0.05.

## 3. Results

### 3.1. Growth

#### 3.1.1. Body Length

Figure 1 shows the mean weekly body length of the three species. The first point on each curve corresponds to the first week after hatching, while the last point represents the final week when individuals reached the adult stage. Shaded bands indicate the standard deviations (SD).

##### *Gryllodes sigillatus* 

This species showed a clear increase in body length at 30 °C (14.93 mm) compared to 28 °C (13.44 mm). The difference is statistically significant (*p* < 0.001), showing that *Gryllodes sigillatus* grows longer in slightly warmer conditions.

##### *Gryllus bimaculatus* 

*Gryllus bimaculatus* shows a small increase at 30 °C (24.69 mm) versus 28 °C (23.76 mm), but this difference was not statistically significant (*p* = 0.188), suggesting that temperature did not have a strong effect on mean final body length in this species.

##### *Teleogryllus lemur* 

This species shows the most striking increase in body length, with an average of 26.06 mm at 30 °C compared to 18.24 mm at 28 °C. This difference was highly significant (*p* < 0.001), indicating that *Teleogryllus lemur* is the most sensitive to the 2 °C increase in temperature.

#### 3.1.2. Body Weight

Figure 2 shows the mean weekly body weight of the three species. The first point on each curve represents the first week after hatching, and the last point corresponds to the final week when individuals reached the adult stage. Shaded bands indicate the standard deviations (SD).

##### *Gryllodes sigillatus* 

This species shows a clear increase in body weight at 30 °C (0.339 g) compared to 28 °C (0.248 g). The difference is statistically significant (*p* < 0.001), suggesting that *Gryllodes sigillatus* performs better under slightly warmer temperatures in terms of body mass accumulation.

##### *Gryllus bimaculatus* 

Body weight also increases at 30 °C (1.049 g) compared to 28 °C (0.891 g), with a significant difference (*p* < 0.001). This result indicates that *Gryllus bimaculatus* benefits from warmer conditions, possibly due to more efficient energy usage or growth rate acceleration.

##### *Teleogryllus lemur* 

This species shows the most pronounced growth difference, with body weight almost doubling at 30 °C (1.204 g) compared to 28 °C (0.534 g). This difference is highly significant (*p* < 0.001), identifying *Teleogryllus lemur* as the most temperature-responsive species in this study.

### 3.2. Survival

The weekly survival curves (Figure 3) illustrate how rearing temperature influenced survivorship across the three cricket species. Survival percentages were calculated based on the number of individuals alive at the end of the trial, from an initial group of 150 individuals per treatment. Confidence intervals (95% CI) in bars were computed as:$$CI\;=\;p\;\pm \;1.96\;\times \sqrt{\frac{p(1-p)}{n}}$$
where *p* is the survival proportion and *n* = 150 (initial count). These intervals provide a range within which the true survival proportion is expected to lie with 95% confidence.

#### 3.2.1. *Gryllodes sigillatus*

Survival differed significantly between temperatures (*p* < 0.001). At 28 °C, survival reached 90%, whereas at 30 °C it decreased to 70%, indicating that *Gryllodes sigillatus* is sensitive to the 2 °C increase in rearing temperature.

#### 3.2.2. *Gryllus bimaculatus*

No significant difference in survival was detected between temperatures (*p* = 0.79). Survival was 72% at 28 °C and 74% at 30 °C, indicating that *Gryllus bimaculatus* is tolerant to the 2 °C increase in rearing temperature.

#### 3.2.3. *Teleogryllus lemur*

Survival differed significantly between temperatures (*p* < 0.05). At 28 °C, survival was 85%, while at 30 °C it declined to 72%, indicating that *Teleogryllus lemur* is sensitive to the 2 °C increase in rearing temperature.

### 3.3. Development Time

The effect of species and temperature on development time was statistically significant (*p* < 0.001, Table 2). At 28 °C, *Gryllodes sigillatus* reached adulthood in 47.82 ± 0.06 days (*n* = 123), whereas at 30 °C, development time increased to 54.07 ± 0.14 days (*n* = 96). In contrast, *Gryllus bimaculatus* developed more rapidly at higher temperatures, averaging 33.56 ± 0.10 days (*n* = 108) at 28 °C and accelerating to 28.80 ± 0.08 days (*n* = 111) at 30 °C. *Teleogryllus lemur* exhibited the longest developmental durations, with 58.69 ± 0.07 days (*n* = 132) at 28 °C, decreasing to 52.24 ± 0.16 days (*n* = 108) at 30 °C.

These results highlight distinct species-specific thermal sensitivities. While *G. bimaculatus* and *T. lemur* showed development acceleration at the higher temperature—consistent with the general ectothermic response—*G. sigillatus* exhibited the opposite trend, with prolonged development at 30 °C. This inverse response suggests that 30 °C may exceed the thermal optimum for *G. sigillatus*, potentially inducing thermal stress that disrupts normal developmental regulation. This species may be better adapted to more moderate temperatures, where metabolic and hormonal pathways critical for growth remain optimally regulated.

The ANOVA results support these observations, revealing significant main effects of both species (F(2, 672) = 31,823.47, *p* < 0.0001) and temperature (F(1, 672) = 474.00, *p* < 0.0001) on total development time. Importantly, a significant species × temperature interaction was detected (F(2, 672) = 2233.31, *p* < 0.0001), indicating that the impact of temperature on development time varied by species.

To assess homogeneity of variances, Levene’s test was conducted and yielded a significant result (*W* = 9.89, *p* = 3.87 × 10^−9^), justifying the use of Welch’s t-tests for subsequent pairwise comparisons. These comparisons revealed significant temperature effects on development time across all species. Specifically, *G. sigillatus* (*t* = −41.26, *p* = 1.28 × 10^−76^), *G. bimaculatus* (*t* = 38.46, *p* = 4.94 × 10^−95^), and *T. lemur* (*t* = 37.12, *p* = 1.66 × 10^−78^) all showed statistically significant changes in development duration between 28 °C and 30 °C confirming that development time at 28 °C differs significantly from 30 °C for all three species.

### 3.4. Biomass Yield

To evaluate the effect of temperature on biomass yield across cricket species, a one-way ANOVA was conducted for each species. Biomass yield was calculated by dividing the total body weights (BW) of surviving adults by the development time specific to each temperature treatment (28 °C and 30 °C). Figure 4 shows the biomass yield by species and temperature.

#### 3.4.1. *Gryllodes sigillatus*

*Gryllodes sigillatus* exhibited a significant difference in biomass yield between temperatures (F(1, 217) = 78.02, *p* < 0.001). Crickets raised at 30 °C achieved a mean biomass yield of 0.81 g/day, compared to 0.76 g/day at 28 °C. The increase in yield reflects both greater final body weight and longer development time under elevated temperature.

#### 3.4.2. *Gryllus bimaculatus*

*Gryllus bimaculatus* displayed a strong temperature response in biomass yield (F(1, 217) = 306.30, *p* < 0.001). Individuals raised at 30 °C produced a mean yield of 4.69 g/day, compared to 3.34 g/day at 28 °C. The faster development time at higher temperature, combined with increased final body weight, resulted in optimal yield performance in this species.

#### 3.4.3. *Teleogryllus lemur*

*Teleogryllus lemur* showed a strong temperature effect on biomass yield (F(1, 238) = 1224.35, *p* < 0.001). Crickets raised at 30 °C yielded 2.89 g/day, compared to 1.40 g/day at 28 °C. This highlights a pronounced temperature-dependent efficiency in converting growth into biomass for this species.

## 4. Discussion

Our study revealed species-specific responses to rearing temperature. *Gryllodes sigillatus* exhibited a moderate increase in development time at 30 °C but maintained high survival rates. *Gryllus bimaculatus* demonstrated accelerated growth and shorter development time at 30 °C without a significant impact on survival. *Teleogryllus lemur*, however, exhibited a significant drop in survival at 30 °C despite increased growth rates. These findings align with previous studies on edible crickets, which report that elevated temperatures can accelerate development but may negatively impact survival and biomass yield [13,21].

Similar research on *G. bimaculatus* found that increasing temperature from 28 °C to 30 °C significantly shortened development time while maintaining stable survival rates [21,24]. However, excessive heat stress beyond 30 °C has been associated with reduced survival and reproductive performance in this species [32]. For *Gryllodes sigillatus*, studies suggest that while temperature influences development speed, this species is relatively temperature-tolerant compared to others [25]. The observed mortality in *T. lemur* at 30 °C highlights a species-specific sensitivity to heat stress, reinforcing the need to tailor farming conditions to each species’ thermal tolerance [21].

Several studies have examined how dietary protein levels interact with temperature to influence cricket growth, feed conversion, and body composition. For instance, *Gryllus madagascarensis* (Walker, 1869) and *G. sigillatus* crickets fed with 21.5% CP showed the highest weight gain and feed efficiency, whereas both lower (13.5%) and higher (28% for *G. madagascarensis* and 25% for *G. sigillatus*) protein levels were associated with reduced performance [13,33]. Similarly, *Gryllus bimaculatus* reached optimal growth and feed conversion ratio at 21.65% CP, while growth performance declined at 24% CP, likely due to metabolic stress from excess protein processing [34].

A limitation of this study is that the protein content of the feeds differed moderately between temperature treatments (18.2% vs. 21.5% CP). Because protein strongly affects growth and feed efficiency, we cannot exclude the possibility that some of the observed differences were due to diet. However, both CP levels fall within the range previously shown to support near-optimal performance in crickets, suggesting that temperature remained the dominant factor.

For *Acheta domesticus*, crickets raised on a 22% CP diet achieved significantly higher body weight and protein content than those on lower protein diets [35]. In another study, both *A. domesticus* and *G. bimaculatus* performed best on plant-based by-product diets containing medium (22.5%) and high (30%) protein levels, with improved survival and growth compared to lower-protein feeds [36]. Moreover, recent studies suggest that reducing dietary protein from 21% to 14–17% in the latter rearing stages can cut feed costs without significantly compromising final biomass yields [37]. Given these findings, our feed treatment falls within the range where cricket performance is typically optimized. Therefore, while slight differences in protein content may contribute to variability, the temperature effects observed in our study are likely the primary driver of performance differences.

Our findings indicate that increasing rearing temperature from 28 °C to 30 °C accelerated development and shortened production cycles, a trend that has been similarly reported in other cricket species under controlled conditions [18,32]. In both *Gryllus bimaculatus* and *Acheta domesticus*, previous studies demonstrated significantly reduced development times at higher temperatures [18], confirming the role of temperature in enhancing metabolic activity and growth [38].

At the same time, our results show that this developmental acceleration came at a cost for certain species, particularly *Teleogryllus lemur*, which exhibited increased mortality at 30 °C. This trade-off between faster growth and reduced survival has been similarly observed in other orthopterans, where mortality increased outside optimal thermal ranges [32,39].

Our data also suggest that the benefits of faster growth may not always outweigh the risks associated with elevated temperatures, especially for species with narrow thermal tolerance. This observation aligns with findings in *Scapsipedus icipe* Hugel & Tanga, 2018, where higher temperatures improved development and fecundity but also led to increased adult mortality [19]. Likewise, while *A. domesticus* showed increased egg production at 30 °C, hatchability sharply declined beyond this threshold [18], illustrating the potential limitations of thermal increases.

For farms equipped with climate control systems, raising temperatures by 2 °C could improve productivity through faster rearing cycles. Our findings support this potential benefit, though they also underscore the need to consider energy costs. Similar trade-offs have been highlighted in global models of *G. bimaculatus*, which showed that although growth improved near 32 °C, survival declined under unregulated high temperatures [40].

In open rearing systems, where artificial heating is unavailable, our study suggests that exposure to fluctuating high temperatures may increase mortality, particularly for heat-sensitive species such as *T. lemur*. In these contexts, strategies such as shade structures, ventilation, and adjusted stocking densities are recommended to mitigate the effects of thermal stress.

Overall, our results reinforce the need to balance developmental benefits with survival risks when manipulating temperature in cricket farming. This conclusion is supported by other studies that similarly found enhanced productivity at moderately elevated temperatures [17,39], but only when mortality remained within manageable limits.

## 5. Conclusions

This study demonstrates that while increasing rearing temperature from 28 °C to 30 °C can accelerate cricket development, it may also impact survival depending on the species. *Gryllus bimaculatus* and *Gryllodes sigillatus* showed promising responses to elevated temperatures, whereas *Teleogryllus lemur* exhibited significant heat sensitivity. These results highlight the need for species-specific farming strategies to balance growth efficiency and survival rates.

For commercial farms with heating systems, the cost–benefit trade-off of raising temperature should be carefully evaluated. In open farming systems, where temperature fluctuations cannot be controlled, species selection and environmental modifications may be more effective strategies than attempting to manipulate temperature.

Given the increasing interest in sustainable cricket farming in Madagascar, future research should explore adaptive farming strategies, such as optimizing housing design, feed composition, and microclimate management, to enhance production efficiency in diverse environmental conditions. Additionally, integrating insect farming with agro-industrial by-products and local feed resources could improve cost-effectiveness and sustainability.

## Figures and Tables

**Figure 1 insects-16-00960-f001:**
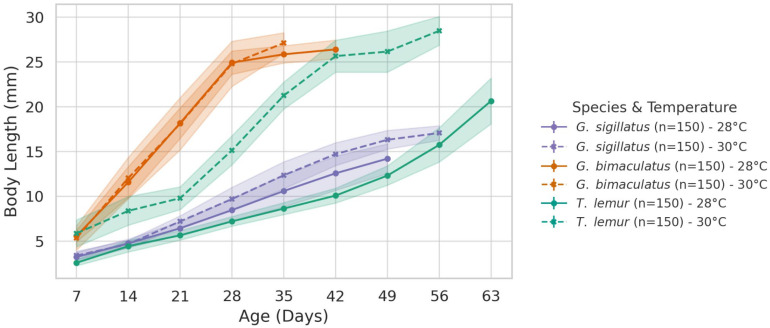
Weekly mean body length of *Gryllus bimaculatus*, *Teleogryllus lemur*, and *Gryllodes sigillatus* reared in temperature 28 and 30 °C.

**Figure 2 insects-16-00960-f002:**
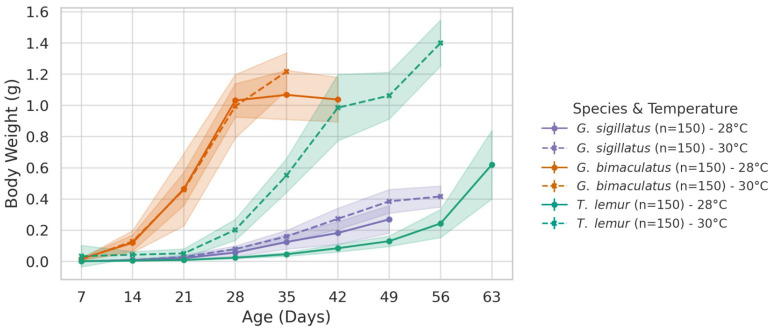
Weekly mean body weight of *Gryllus bimaculatus*, *Teleogryllus lemur*, and *Gryllodes sigillatus* reared in temperature 28 and 30 °C.

**Figure 3 insects-16-00960-f003:**
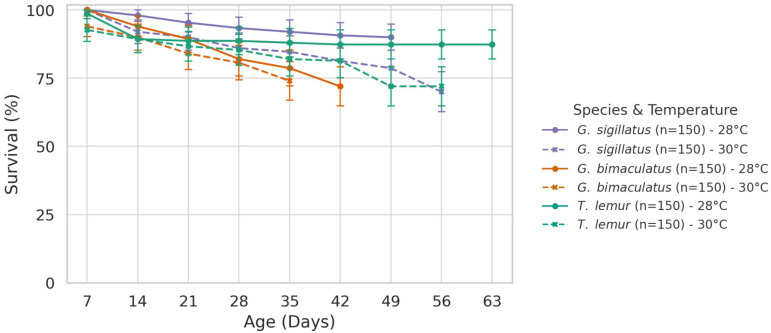
Weekly survival rate curves of *Gryllus bimaculatus*, *Teleogryllus lemur*, and *Gryllodes sigillatus* reared in temperature 28 and 30 °C.

**Figure 4 insects-16-00960-f004:**
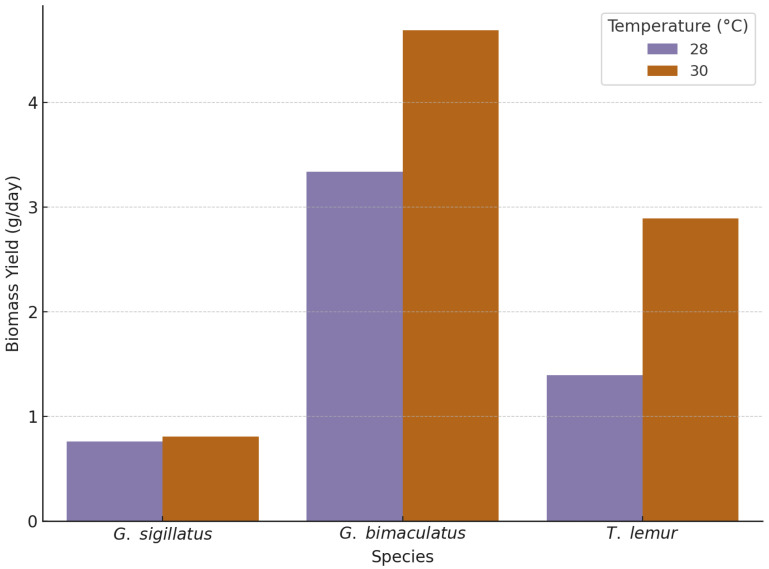
Biomass yield of *Gryllus bimaculatus*, *Teleogryllus lemur*, and *Gryllodes sigillatus* reared in temperature 28 and 30 °C.

**Table 1 insects-16-00960-t001:** Cricket species used in the study, with corresponding collection locations, geographic coordinates, and habitat type in Madagascar.

Species	Location	Coordinates (Lat, Long)	Habitat Information
*Gryllus bimaculatus*	Fort Dauphin (southeastern region of Madagascar)	−24.9615, 46.9839	Grassland, farming areas
*Teleogryllus lemur*	Morondava (western part of Madagascar)	−20.2930, 44.2855	Farming areas, grasslands, forest clearings, riversides
*Gryllodes sigillatus*	Mahajanga (northwestern region of Madagascar)	−15.6526, 46.3793	Synanthropic, villages, farming areas

**Table 2 insects-16-00960-t002:** Development time (Mean ± se days) of *Gryllus bimaculatus*; *Gryllodes sigillatus* and *Teleogryllus lemur* at temperature 28 and 30 °C.

Species	Temperature	Mean	Development Time
*G. sigillatus*	28	47.82	47.82 ± 0.06 days (*n* = 123)
*G. sigillatus*	30	54.07	54.07 ± 0.14 days (*n* = 96)
*G. bimaculatus*	28	33.55	33.56 ± 0.10 days (*n* = 108)
*G. bimaculatus*	30	28.80	28.80 ± 0.08 days (*n* = 111)
*T. lemur*	28	58.68	58.69 ± 0.07 days (*n* = 132)
*T. lemur*	30	52.24	52.24 ± 0.16 days (*n* = 108)

## Data Availability

All data supporting the findings of this study are openly available in the GitHub repository “Enhancing Cricket Farming Efficiency” at https://github.com/ValRian/Enhancing-Cricket-Farming-Efficiency (accessed on 30 June 2025).

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
