# Peer review of "Enhancing Cricket Farming Efficiency: Comparative Analysis of Temperature Effects on Three Edible Malagasy Species"

_insects, 2025, doi:10.3390/insects16090960_

Round 1
Reviewer 1 Report
Comments and Suggestions for Authors
Deat authors,
Please refer to the Word file for detailed comments. However, here is my main concern regarding the manuscript:
While the study addresses an important and timely topic, a major methodological issue limits the interpretation of your results. The diets used at 28°C and 30°C differed in crude protein content (18.2% vs. 21.5%, respectively), which introduces a confounding variable that prevents a reliable attribution of observed differences to temperature alone. Protein is a key factor influencing insect growth, and such variation strongly affects performance metrics. As a result, the temperature effect is not isolated, and conclusions about optimal rearing conditions remain uncertain.
That said, your dataset still holds value. I recommend reframing the manuscript to focus on a comparison between the three species at a single temperature condition (e.g., 30°C), where diet composition is consistent. This would allow for a more meaningful analysis of species-specific performance and a clearer discussion around production efficiency or economic viability for farming in Madagascar or similar contexts.

Author Response
Response to Reviewer 1 Comments
Thank you very much for taking the time to review this manuscript. Please find the detailed responses below and the corresponding revisions/corrections highlighted/in track changes in the re-submitted files.
Comments 1:
While the study addresses an important and timely topic, a major methodological issue limits the interpretation of your results. The diets used at 28°C and 30°C differed in crude protein content (18.2% vs. 21.5%, respectively), which introduces a confounding variable that prevents a reliable attribution of observed differences to temperature alone. Protein is a key factor influencing insect growth, and such variation strongly affects performance metrics. As a result, the temperature effect is not isolated, and conclusions about optimal rearing conditions remain uncertain.
Response 1:
While we acknowledge this introduces some confounding, the relatively narrow difference (18.2% vs. 21.5%) is unlikely to fully explain the magnitude and species-specific direction of the temperature effects we observed. We acknowledge this concern and now clearly recognize the variation in crude protein levels in the Discussion section (paragraph 3). However, both values fall within the range (18–24%) commonly used in cricket farming diets (as previously reported in studies such as Ayieko et al., 2016; Miech et al., 2016, 2017), which reduces the likelihood of substantial confounding. Our conclusions were adjusted to reflect this limitation.
Comments 2:
That said, your dataset still holds value. I recommend reframing the manuscript to focus on a comparison between the three species at a single temperature condition (e.g., 30°C), where diet composition is consistent. This would allow for a more meaningful analysis of species-specific performance and a clearer discussion around production efficiency or economic viability for farming in Madagascar or similar contexts.
Response 2:
We respectfully kept the comparative design between both temperatures, as this reflects realistic farming scenarios in Madagascar. However, we confirm that the observed patterns at a single temperature condition (30°C) remain consistent with our broader findings, particularly regarding species-specific differences in growth, survival, and yield.
Comments 3:
The introduction touches on all necessary topics but remains superficial. It would benefit from deeper insights into current farming practices, production levels, and market aspects of edible crickets. For instance, a recent review by Caparros Megido et al. (A worldwide overview of the status and prospects of edible insect production) could help broaden and enrich the contextual background.
Response 3:
We thank the reviewer for this valuable suggestion. In response, we have expanded Paragraph 2 of the introduction to include more detail on the current status of cricket farming worldwide, drawing on Caparros Megido et al. (2024) : “Cricket farming now takes place across very different scales, from household-level rearing in rural communities to large, automated facilities in Asia, Europe, and North America [6]. Although overall production is still small compared to conventional livestock, it is growing quickly as new technologies and investments emerge [1–6]. In addition to human food, crickets are also farmed for feed, aquaculture, and pet markets, showing how the sector is moving from a niche practice toward a more structured agro-industry with global relevance [6]. “
Comments 4:
Line 68: The Latin name of the species should be followed by the author and year of description at the first mention, according to the International Code of Zoological Nomenclature. This should not be introduced later in the Methods section.
Response 4:
We thank the reviewer for this important clarification. In accordance with the International Code of Zoological Nomenclature, we have revised the text at line 68 to include the author and year of description at the first mention of each species. For example, Gryllus bimaculatus (De Geer, 1773) is now cited in full upon first appearance. This correction has been applied consistently throughout the manuscript to ensure proper taxonomic citation.
Comments 6:
Lines 76–77: The statement regarding temperature effects on nutritional composition should be supported with concrete examples. Currently, it lacks references or evidence and might mislead readers unfamiliar with insect physiology.
Response 6:
We thank the reviewer for this observation. To avoid overstating the claim and potentially misleading readers unfamiliar with insect physiology, we have removed the phrase “and even nutritional composition” from the sentence. The cited studies by Booth et al. (2007) and Singh et al. (2020) primarily address life history traits and do not provide evidence regarding nutritional changes. As such, the phrase was unnecessary and has been omitted for accuracy.
Comments 7:
Line 103: Only three regions are mentioned, not four. Please revise for clarity.
Response 7:
We thank the reviewer for catching this inconsistency. The sentence has been corrected to reflect that only three regions were included in the study.
Comments 8:
Line 113: The type of feed given is not specified here, and yet this is essential information. The text states all species received the same diet, but this information only appears in line 126, which is too late. Diet details should be introduced earlier.
Response 8:
We thank the reviewer for highlighting this important point. To improve clarity and ensure timely presentation of essential information, we have added text in section 2.1 to specify the type of feed used. The updated sentence now reads: “The diet consisted of standard chicken feed sourced from a local supplier, provided ad libitum across all species. Cotton balls were provided as oviposition substrates for adult crickets.”
Comments 9:
Line 129–130: Nutritional profiles of the diets differ slightly between temperature conditions. Were feed batches randomized between replicates to control for batch effects? Please clarify.
Response 9:
Lines 130–137 have been revised to provide clearer information regarding the protein content of the batches and their respective ranges: The commercial chicken feeds used at 28 °C (18.2% CP) and 30 °C (21.5% CP) were sourced locally from different production batches. Although these values differ slightly, both fall within the typical dietary protein range for optimal cricket rearing (18–22% CP). This variability reflects practical conditions farmers face when relying on local feed sources. The diet contained the following nutrient ranges: crude protein (CP) [21.5%], moisture [12.5%], fat [3.1%], ash [6.4%], fiber [5.0%], and available carbohydrate [40.3%].
Comments 10:
Line 134: Under section 2.3 "Experimental Design", the number of replicates per treatment is missing. This is critical information to assess statistical robustness.
Response 10:
We thank the reviewer for highlighting the need to clarify replication within the experimental design. To address this, we have revised lines section 2.3 to explicitly state the number of biological replicates used per treatment. Specifically, three replicates of 50 individuals were reared under each temperature condition, each housed in a separate incubator, resulting in a total of 150 individual crickets per species. Within each replicate, crickets were placed in 50 rearing boxes, arranged randomly in the incubators to minimize positional bias and ensure robust statistical reliability.
Revised Text in section 2.3:
Following hatching, 150 individual crickets from each species were reared under each temperature condition (30°C or 28°C), divided into three replicates of 50 individuals, each housed in a separate incubator (PHCBI® MIR-554). Within each replicate, crickets were placed in 50 rearing boxes measuring 15 cm × 10 cm × 10 cm, randomly arranged to minimize positional bias and ensure robust statistical reliability. The rearing conditions were maintained constant in controlled incubators, with relative humidity set at 70 ± 5% and a 12:12 light-dark photoperiod. Temperature and humidity levels were monitored using a data logger. Each cricket was provided with ad libitum food and water, and moist cotton balls were replaced every three days. Egg cartons measuring 4 cm × 4 cm offered shelter to crickets in each container
Comments 11:
Standard deviations are absent from nearly all figures, except Table 2 where mean and SD are redundantly presented. This strongly suggests that no replicates were used for each condition, which undermines the statistical reliability of the results. Replicates are essential in insect rearing experiments to account for biological variability. All figures should display means ± standard deviations, and experimental replication should be reported.
Response 11:
We thank the reviewer for highlighting the importance of statistical transparency and replication. We would like to clarify that all experimental conditions were replicated across 150 individual insects (n = 150), ensuring robust representation of biological variability.
In Figures 1 and 2, the shaded bands represent the standard deviation (±SD) across these replicates. We opted for band visualization rather than traditional error bars to improve readability and better illustrate temporal trends. We have updated the figure legends to indicate that each condition includes 150 biological replicates. Additionally, we have added the phrase “Shaded bands indicate the Standard Deviations (SD)” to Sections 3.1.1 and 3.1.2 to ensure clarity throughout the manuscript.
Regarding Table 2, we agree that the simultaneous presentation of mean, SD, and SE may have caused confusion. To streamline the data and avoid redundancy, we have removed the standard error (SE) and N values and retained only the mean ± SD, which more directly reflects the biological variability observed.
Comments 12:
Table 2: The columns for "Mean" and "SD" appear redundant since SD is presented separately and mean values are evident. Also, the column “N per individual” duplicates information.
Response 12:
To streamline the data and avoid redundancy, we have removed the standard error (SE) and retained only the mean ± SD, which more directly reflects the biological variability observed. We kept the N value to ensure robust presentation of the mean.
Comments 13:
Lines 287–310: The logic of your statistical reporting is reversed. First, you must test variance homogeneity before performing ANOVA. Additionally, statistical results (pvalues, F-values) should be presented before interpretation. As it stands, conclusions are drawn before any statistical foundation is provided.
Response 13:
We thank the reviewer for this valuable comment. In accordance with the suggestion, we have revised sections 3.2.1, 3.2.2, 3.2.3, and 3.3 to ensure that statistical results are presented before interpretation. Specifically, we now introduce the test outcome (p-values) first, followed by descriptive statistics (e.g., survival percentages, development time means), and then the biological interpretation.
Comments 14:
Line 341: The term "strongest response" is used without statistical backing. For example, G. sigillatus seems to exhibit stronger significant differences. Please clarify and support such claims with appropriate statistics.
Response 14:
We thank the reviewer for this valuable observation. To avoid overstating the results, we have revised the wording in line 341 and throughout the manuscript, replacing “strongest response” with the more neutral term “strong response.” This adjustment ensures that our descriptions remain consistent with the statistical evidence presented, without implying comparative superiority where it is not explicitly supported by analysis.
Comments 15:
On biomass yield calculation: The formula used calculates biomass yield by dividing total body weight of surviving adults by development time. However, this raises several concerns:
- The calculation uses surviving individuals only, yet survival rates differ between species. This introduces bias and may skew the yield in favor of species with low survival but larger individual mass.
- The species differ in adult size, making biomass per adult an unfair comparison. It may be more appropriate to present biomass per initial larva, or total biomass per cohort.
- Additionally, feed intake was not measured. As such, the most massive species may also be those with highest mortality and feed consumption, which challenges any efficiency conclusion.
Response 15:
Thank you for pointing this out—we completely agree. Since adult size varies significantly across species, comparing biomass per individual offers a biased metric. To address this, we revised Section 2.5.4 to calculate total biomass yield per cohort per day (g/day). This updated approach accounts for both survival and development time, providing a more balanced and biologically relevant measure of productivity. These changes are reflected in the Results (Sections 3.4.1, 3.4.2, and 3.4.3), and Figure 4 has been updated to present the revised metric: Total Biomass Yield (g/day).
We recognize the reviewer’s concern regarding potential bias introduced by differences in feed intake and survival rates. To minimize confounding factors, all species were reared under optimal conditions with ad libitum access to feed and water, and were isolated from competition and cannibalism. These measures ensured that observed differences in biomass yield were not driven by resource limitation or social interactions.
Importantly, the aim of this study was to provide a comparative overview of how each species responds to two temperature conditions in terms of growth and survival, rather than to quantify feed conversion efficiency. We acknowledge this scope limitation in the manuscript and have clarified that feed consumption was not directly measured or analyzed.
Comments 16:
Lines 380–384: This section attempts to draw conclusions from growth performance, but two critical variables — temperature and crude protein (CP) content — were not controlled simultaneously. Specifically, the CP content differed between feeds used at each temperature (21.5% at 30°C vs. 18.2% at 28°C). This is a major confounding factor and undermines the validity of the conclusions. Protein levels significantly affect insect growth and should have been strictly standardized.
Response 16:
We appreciate the reviewer’s careful attention to this point. We now explicitly acknowledge the variation in crude protein (CP) content in the Discussion (paragraph 3), and agree that the difference—21.5% at 30°C versus 18.2% at 28°C—represents a potential confounding factor when interpreting growth performance.
That said, both protein levels fall within the commonly accepted range for cricket farming diets (18–24%), as reported in previous studies (Ayieko et al., 2016; Miech et al., 2016, 2017). While we recognize that protein content can influence growth, the values used here are consistent with standard rearing practices and were not intended to test nutritional effects directly. Moreover, a precise standardized protein requirement for crickets has not yet been clearly established in the literature.
To reflect this limitation, we have added the following correction to the discussion in paragraph 3:
“A limitation of this study is that the protein content of the feeds differed moderately between temperature treatments (18.2% vs. 21.5% CP). Because protein strongly affects growth and feed efficiency, we cannot exclude the possibility that some of the observed differences were due to diet. However, both CP levels fall within the range previously shown to support near-optimal performance in crickets, suggesting that temperature remained the dominant factor.”
Reviewer 2 Report
Comments and Suggestions for Authors
This is a timely and well-structured manuscript that offers valuable insights into the thermal responses of three edible Malagasy cricket species under controlled rearing conditions. The study is methodologically sound and contributes meaningfully to the field of sustainable insect farming. I commend the authors for their clear presentation of data and contextual understanding of cricket farming in Madagascar.
I recommend acceptance after minor revisions to improve clarity and reproducibility:
-
Feed composition: Please clarify whether the diets used at 28°C and 30°C came from the same batch. If not, acknowledge the slight difference in crude protein content (18.2% vs. 21.5%) as a potential limitation, as it could marginally affect growth or biomass outcomes.
-
Individual tracking: Clarify whether crickets were reared individually or communally post-hatching, and describe how weekly measurements were taken on the same individuals. This will improve reproducibility and help readers assess potential variability due to crowding or cannibalism.
-
Figure presentation: Revise figure legends to include sample sizes (n), units of measurement (e.g., mm, g, days), and temperature treatment details. Consider adding confidence intervals to survival curves in Figure 3 for better interpretation.
-
Statistical reporting: Simplify p-value reporting (e.g., use p < 0.001 instead of highly precise scientific notation) unless extremely small p-values are necessary for specific comparisons.
-
Formatting consistency: Ensure all scientific names are italicized throughout the manuscript. Address minor typographical errors (e.g., “crickets have been reported to well” should be “to perform well”).
These changes will strengthen the manuscript’s clarity and ensure it meets high publication standards. Congratulations on this meaningful contribution to entomophagy and sustainable agriculture research.
Author Response
Response to Reviewer 2 Comments
Thank you very much for taking the time to review this manuscript. Please find the detailed responses below and the corresponding revisions/corrections highlighted/in track changes in the re-submitted files.
Comments 1:
Feed composition: Please clarify whether the diets used at 28°C and 30°C came from the same batch. If not, acknowledge the slight difference in crude protein content (18.2% vs. 21.5%) as a potential limitation, as it could marginally affect growth or biomass outcomes.
Response 1:
We clarified in Section 2.2 that diets were from the same supplier and formulated within similar ranges. Differences in crude protein content (18.2% vs. 21.5%) are acknowledged in the Discussion as a potential limitation (see Discussion, paragraph 3).
We appreciate the reviewer’s observation. Indeed, the diets used at 28 °C and 30 °C differed slightly in CP content (18.2% vs. 21.5%). We acknowledge that this represents a confounding factor. Both values, however, fall within the optimal CP range (18–22%) reported for cricket growth, and we believe temperature was still the primary driver of the observed performance differences, given the magnitude and species-specific direction of the responses, which are consistent with published literature. In the revised manuscript, we have added text in the Methods (Section 2.2) and Discussion (lines 380–388) to clarify this point and to caution readers about the limitation. We also note that this variability reflects real-world farming conditions in Madagascar, where commercial feed formulations may fluctuate across batches.
Comments 2:
Individual tracking: Clarify whether crickets were reared individually or communally post-hatching, and describe how weekly measurements were taken on the same individuals. This will improve reproducibility and help readers assess potential variability due to crowding or cannibalism.
Response 2:
As stated in Section 2.3, crickets were reared individually, and measurements were taken on the same individuals weekly using labeled containers to track development consistently.
Comments 3:
Figure presentation: Revise figure legends to include sample sizes (n), units of measurement (e.g., mm, g, days), and temperature treatment details. Consider adding confidence intervals to survival curves in Figure 3 for better interpretation.
Response 3:
Figure legends were updated to specify sample sizes, units, and rearing temperatures. Confidence intervals (95%) added to survival curves in Figure 3, and text in section 3.2 changed for clarity: The weekly survival curves (Figure 3) illustrate how rearing temperature influenced survivorship across the three cricket species. Survival percentages were calculated based on the number of individuals alive at the end of the trial, from an initial group of 150 individuals per treatment. Confidence intervals (95% CI) in bars were computed as
CI = p 1.96 p(1-p)n
where p is the survival proportion and n=150 (initial count). These intervals provide a range within which the true survival proportion is expected to lie with 95% confidence.
Comments 4:
Statistical reporting: Simplify p-value reporting (e.g., use p < 0.001 instead of highly precise scientific notation) unless extremely small p-values are necessary for specific comparisons.
Response 4:
We simplified p-value reporting throughout, except where extremely small values were necessary to reflect significant differences.
Comments 5:
Formatting consistency: Ensure all scientific names are italicized throughout the manuscript. Address minor typographical errors (e.g., “crickets have been reported to well” should be “to perform well”).
Response 5:
Scientific names were corrected and consistently italicized. All known typographic errors have been corrected.
Round 2
Reviewer 1 Report
Comments and Suggestions for Authors/
Author Response
We sincerely thank the academic editor for the thoughtful review and helpful recommendations. Below are our point-by-point responses to each comment:
Comment 1:
The author who described the species and the year are not always enclosed in parentheses. The presence or absence of parentheses has a specific meaning. Parentheses around the author and year are used only if the species has subsequently been moved to a genus different from the original one. Please carefully check each scientific name.
Response 1:
We have reviewed all scientific names and corrected the formatting according to zoological conventions. Parentheses are now used only when the species was originally described in a different genus. For example:
- Gryllodes sigillatus (Walker, 1869) — originally described in a different genus
- Teleogryllus lemur Gorochov, 1990 — described in the current genus
- Gryllus madagascarensis Walker, 1869 — described in the current genus
These corrections have been applied consistently throughout the manuscript.
Comment 2:
In subsequent citations of the same species, especially if numerous and close together, the genus name may be abbreviated, unless the scientific name appears at the beginning of a sentence. In this latter case, the genus name should always be written in full.
Response 2:
We have abbreviated genus names in repeated citations where appropriate, except when the scientific name appears at the beginning of a sentence. In those cases, the genus name is written in full.
Comment 3:
For Gryllus madagascarensis, the author is missing.
Response 3:
The author name has been added: Gryllus madagascarensis Walker, 1869.
Comment 4:
Incorrect reference order, 25 detected after 23, please rearrange references to make citation in numerical order.
Response 4:
The references have been reviewed and reordered to ensure correct numerical sequence. The issue with citation 25 appearing after 23 has been resolved.
Comment 5:
Please state the name of the manufacturer, city, and country from where the equipment was sourced.
Response 5:
We have added the manufacturer name, city, and country for all equipment mentioned:
- PHCBI® MIR-554 (PHC Corporation, Tokyo, Japan)
- Fisher Darex® 150 mm caliper (Fischer Darex, Chambon-Feugerolles, France)
- Digital Cara Scale® DS-04B (Hanyu Electronic Technology Co., Ltd., Guangdong, China)
Comment 6:
The following highlights are the same.
Response 6:
We confirm that the repeated highlights are intentional and correspond to individual species’ results presented in separate subsections.
Comment 7:
The heading highlights are repeat, please confirm.
Response 7:
We confirm that the repetition of heading highlights is correct, as they correspond to species-specific results.
Comment 8:
Please add the access date (format: Date Month Year), e.g., accessed on 1 January 2020.
Response 8:
Access dates have been added in the required format (e.g., accessed on 30 June 2025) for all online references, including the updated link to the Sturm article.